# Exploiting Semantic Relations for Glass Surface Detection

**Jiaying Lin**[*]    **Yuen-Hei Yeung**[*]    **Rynson W. H. Lau**[†]
Department of Computer Science
City University of Hong Kong
{jiayinlin5-c, yh.y}@my.cityu.edu.hk, Rynson.Lau@cityu.edu.hk

## Abstract

Glass surfaces are omnipresent in our daily lives and often go unnoticed by the majority of us. While humans are generally able to infer their locations and thus avoid collisions, it can be difficult for current object detection systems to handle them due to the transparent nature of glass surfaces. Previous methods approached the problem by extracting global context information to obtain priors such as object boundaries and reflections. However, their performances cannot be guaranteed when these deterministic features are not available. We observe that humans often reason through the semantic context of the environment, which offers insights into the categories of and proximity between entities that are expected to appear in the surrounding. For example, the odds of the co-occurrence of glass windows with walls and curtains are generally higher than that with other objects, such as cars and trees, which have relatively less semantic relevance. Based on this observation, we propose a model named *Glass Semantic Network ('GlassSemNet') that integrates the contextual relationship of the scenes for glass surface detection with two novel modules: (1) Scene Aware Activation (SAA) Module to adaptively filter critical channels with respect to spatial and semantic features, and (2) Context Correlation Attention (CCA) Module to progressively learn the contextual correlations among objects both spatially and semantically. In addition, we propose a large-scale glass surface detection dataset named Glass Surface Detection - Semantics ('GSD-S'), which contains 4,519 real-world RGB glass surface images from diverse real-world scenes with detailed annotations for both glass surface detection and semantic segmentation. Experimental results show that our model outperforms state-of-the-art works, especially with 42.6% MAE improvement on our proposed GSD-S dataset. Code, dataset, and models are available at* `https: // jiaying. link/ neurips2022-gsds/`

## 1   Introduction

Glass surfaces, including glass doors, windows, and walls of modern architecture, are becoming prevalent in our daily lives. Due to the ambiguity of the transparency property, the autonomous systems of contemporary works typically lack the ability to identify glass surfaces. With such characteristics, the peripheral environment displayed by the glass surface only contains opaque objects and scenes from the surroundings. These result in a myriad of potential dangers caused by the impairment of the existing object detection models in handling glass surfaces, as manifested in previous works [1, 2]. Consequently, it brings a pressing need for a better glass detection model. Existing methods have explored many characteristics of glass surfaces, including context [3],

---

[*]Joint first authors.

[†]Corresponding author.

36th Conference on Neural Information Processing Systems (NeurIPS 2022).

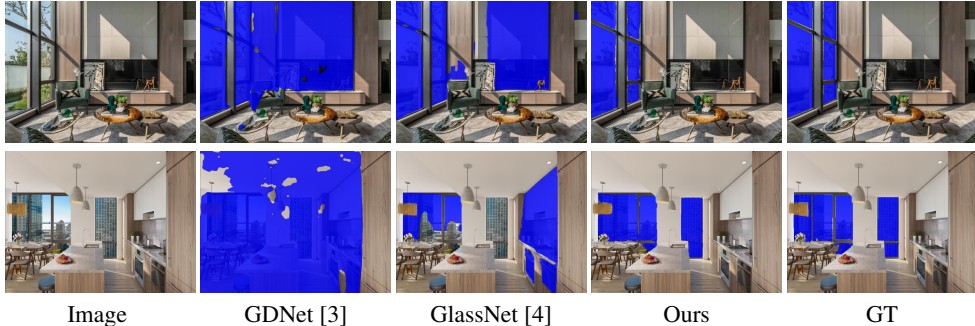

| Image | GDNet [3] | GlassNet [4] | Ours | GT |

Figure 1: Existing methods [3, 4] may fail when explicit physical cues (e.g., boundaries and reflections) are unreliable. In the top row, the center region has glass-like boundaries, and existing methods wrongly predict this region as glass. In the bottom row, as the smaller window on the right does not have obvious reflection, it can be perplexing to detect. Our model can accurately detect glass surfaces in both situations, which are not distracted by regions with glass-like boundaries through learning semantics. It generalizes well even to glass regions without obvious reflection.

boundary [4], reflection [4] and polarization [5]. Although these methods generally perform well, when the assumptions that these models make do not exist, the detection ability of these models is significantly impeded. In addition, these methods only exploit the pixel-level binary labels for glass surfaces, which introduces a bottleneck for the model on knowledge learning, as opposed to other models with multi-class learning that encourage knowledge acquisition through a more diverse pool of knowledge.

In this paper, we observe that there is often a correlated occurrence of glass surfaces with some surrounding objects in the scene. For example, glass surfaces of 'windows' tend to co-occur with 'curtains' (or 'blinds') and 'glass doors' with 'walls'. This prompts us to reconsider the problem from a new perspective by incorporating knowledge on top of superficial features. Hence, we propose to focus more on semantic context learning. Studies in cognitive neuroscience [6, 7, 8, 9] demonstrate that surrounding objects can offer an effective derivation of contextual information. In addition, applications of contextual information have had many proven successful impacts. Examples tasks include salient object detection [10], domain adaptation [11], network pruning [12], image style transfer [13], and image-text retrieval [14]. Based on these and our findings, we propose to learn and incorporate the relationship between glass surfaces and their surrounding semantic context for glass surface detection.

Figure 1 shows that existing methods are often misled by obscure scenes wherein explicit physical cues such as boundaries and reflections are missing. To address these issues, we propose a novel model exploiting semantic relations for glass surface detection. GlassSemNet adopts the encoder-decoder architecture. The encoder component is supported by two backbones: 1) a SegFormer network [15] for comprehensive spatial context learning, and 2) a ResNet50 network [16] for semantic relationship learning. The semantic backbone (DeepLabV3P-ResNet50) is first pre-trained with regular segmentation modeling to comprehend scene context reference and object relationships in real-world settings. The backbone features from the DeepLabV3+ classifier are projected to semantic encodings that are fed into downstream modules to serve as contextual hints for subsequent information extraction. Specifically, two modules are proposed to achieve the ontology learning objectives to assist glass surface detection: 1) the Scene Aware Activation (SAA) Module to guide contextual feature modeling, and 2) the Context Correlation Attention (CCA) Module to associate spatial context and semantic meanings of objects in the environment. Inspired by SENet [17], the SAA Module consists of two feature selection pathways that respectively assimilate the information concerning object locations and object categorical connotations. The CCA Module adopts the Transformer block [18] to conduct attention modeling between the extracted backbone features.

Besides, we notice that although Mei *et al.* [3] and Lin *et al.* [4] both propose datasets for glass surface detection, these datasets do not contain semantic data (e.g., semantic labels of different objects around glass surfaces) to model the spatial context and high-level scene context. To further the research on glass surface detection, we propose a new large-scale challenging semantic-aware

glass surface dataset (GSD-S) with ground truth semantic labels, not only limited to the binary masks of glass surfaces. Our dataset consists of 4,519 images collected from various scenes. It is larger than those proposed by Mei *et al.* [3] (3,900 images) and Lin *et al.* [4] (4,102 images), and can largely facilitate research in this area. Extensive experiments and evaluations on all three datasets were conducted to validate the performance of our model.

Our contributions can be summarized as follows:

- We propose a strategy to apply semantic relationship modeling to cognitively infer the correlation between glass objects and everyday objects for glass surface detection.

- We present two novel deep learning modules to capture long-range spatial and implicit semantic dependencies, with results being substantiated by thorough studies.

- We have built a large-scale dataset, which has complex and challenging scenes with semantic contexts. It can serve as a benchmark for performance validation on future models.

- We have conducted extensive experiments to evaluate our model's robustness and show that it outperforms state-of-the-art methods on glass surface detection.

## 2   Related Work

**Transparent Object Detection.** Transparent Object Detection aims to identify glass-made objects, specifically with bounded shapes such as glasses and glass bottles; occasionally, the task also accommodates window panels. Existing works approached this task by leveraging the bounded shapes to localize the position of prospective transparent objects. The methods range from as simple as adopting an encoder-decoder structure to extract boundary [19, 20, 21] and surface normal [22]. Light polarization was utilized to capture the rotation of light waves from a Physics perspective [23] and multi-view stereo images to generate depth maps that further outline the object shape information [23]. However, glass panels with flat surfaces usually do not possess the boundary characteristic, which induces an even more challenging obstacle for detection models.

**Context-Aware Detection.** Context-aware methods tackle the limited receptive field bottlenecks of convolutional kernels. Most works employed auxiliary operations such as dilation [24, 25] and pooling [26] to enlarge the receptive field such that it gains more global contextual information. More specific solutions that are tailored for various types of surface detection also follow this fashion of contextual learning, such as for mirror surface [27, 28], and glass surface [3, 4]. Nevertheless, these methods are yet another feature aggregation strategy that leaves behind the implicit reasoning embedded in the network learning and rarely exploits the explicit semantic relationship.

**Attention-based Detection.** Attention mechanism from [29] enables the modeling to be more specific and oriented towards meaningful context. Early works used matrix formulation to construct such attention purpose [26, 30, 31]. A boost in performance is nurtured by [32], which actualized the 'transformer' concept in Computer Vision tasks. Nonetheless, it is still in the form of spatial context, which relies on semantic feature extraction from each local patch. [33, 34] proposed novel strategies to instead focus on semantic context to study the relationships between objects and scenes for semantic meanings, which served as an inspiration for our work to utilize semantic dependency. Subsequent work [35] swiftly merged the 'transformer' concept and semantic category embedding in transparent object detection. However, the model is constrained to model relationships between a few types of transparent objects, e.g., eyeglasses, bowls, freezers, and windows. This method completely ignores and disposes of the potential of semantic meanings between object categories and scenic information; we aim to fill the gap by emphasizing contextual relationships.

## 3   Proposed Dataset

We acknowledge that there exist numerous datasets [36, 37] that are dedicated to semantic segmentation tasks. Since most of them are augmented for everyday objects and thus have a general classification purpose, more refined labelings would be required. Consequently, a rectified dataset that caters explicitly to glass surface detection concerning semantic context is still in need. On the other hand, [3] and [4] are among the earliest teams who pioneered the 'glass detection' studies and contributed large-scale glass datasets. While half of [4]'s GSD dataset was assembled from

Table 1: Composition of our proposed GSD-S dataset. We collect glass images from four existing RGB image datasets with semantic annotations. Note that these datasets initially lack refined annotations of ground truth glass surface masks. When constructing our dataset, we re-labeled the GT masks for the glass surfaces.

| Dataset | Whole | Train | Test |
|---|---|---|---|
| SUN RGB-D [38] | 1,203 | 920 | 283 |
| 2D-3D-Semantics [39] | 600 | 488 | 112 |
| Matterport3D [2] | 1,206 | 992 | 213 |
| COCO-Stuff [37] | 1,511 | 1,511 | N/A |
| Total | 4,519 | 3,911 | 608 |

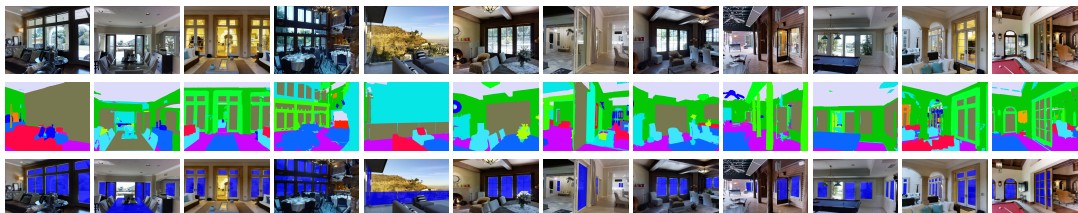

Figure 2: Preview of the GSD-S Dataset. Triplets of (RGB images, semantic maps, overlaid GT mask of glass surfaces).

existing semantic segmentation datasets (e.g. [36, 37]), [3]'s GDD dataset was constructed from manual collection and organization with only ground truth glass masks. We herewith propose a new Semantic-Aware Glass Surface Detection (GSD-S) dataset, which is accompanied by polished ground truth 'glass surface' masks along with semantic segmentation ground truths, with the hope that this contributes to further extensions.

**Dataset Composition.** The GSD-S dataset is scrupulously organized from existing semantic segmentation datasets [40] with the corresponding ground truth annotation carefully refined since the glass mask labelings in original versions were in general inconsistent. Examples of ambiguous mask labeling include glass areas being segmented into different categories of surrounding objects due to the glass surface transparency; and glass surfaces that were ignored and treated as part of the larger subject, such as cupboard (glass door), car (glass window), table (glass table). We processed 4,519 images, with 3,911 training images and 608 testing images altogether. The split between training and testing sets strictly follows that of the original datasets whenever possible (subset from Matterport3D [40] was randomly sampled). Table 1 gives an overview of the mentioned distribution. Figure 3 displays the object category distributions of 43 classes in our GSD-S dataset. Both training and testing sets follow similar distributions. The area ratio of GSD-S is distinctively small compared to the other datasets due to the consideration that more semantic context can be included, which allows comprehensive scene context relationship modeling.

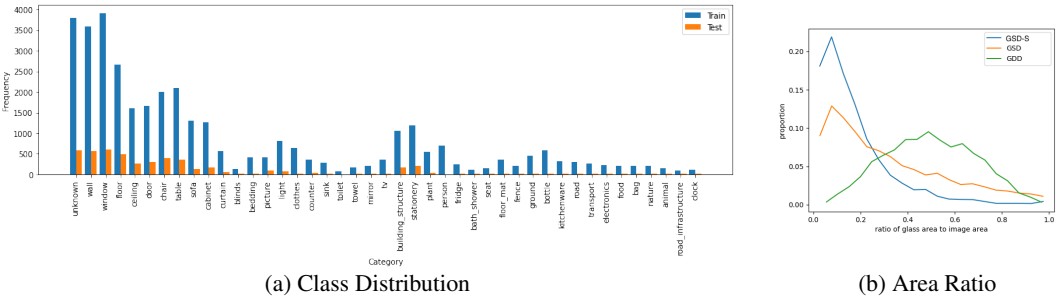

(a) Class Distribution                                    (b) Area Ratio

Figure 3: Dataset Statistics: (a) class distribution, and (b) area ratio.

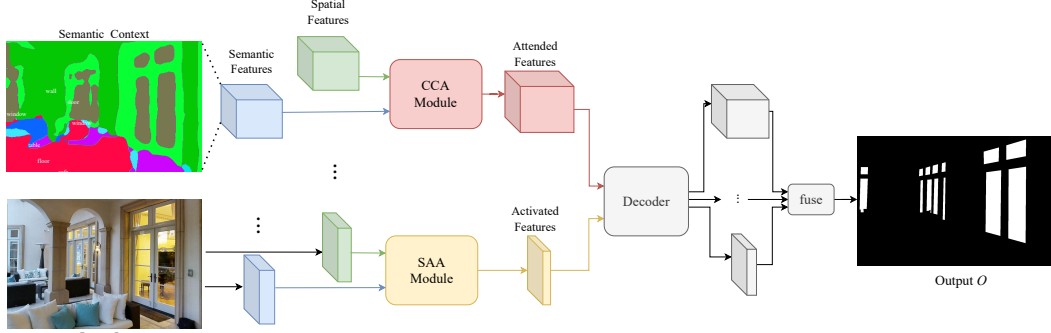

Figure 4: Model Architecture. We first feed the input image into two backbones to capture semantic knowledge, and spatial location features separately. Together with the semantic encodings, low-level features first get selectively activated by the SAA Module with respect to the decoupled features. The CCA Module is placed at a higher level to learn the relationships between contextual meanings and locations of objects. Features from multiple stages are aggregated by the UPerNet decoder to produce the output map, along with the intermediate feature maps for supervision.

# 4    Proposed Method

Figure 4 illustrates the proposed model's architecture. The input image is first fed into two backbone networks for spatial and semantic feature extraction. To bridge up upstream and downstream module features, we introduce the semantic encodings ($f_{sem\_encod} \in \mathbb{R}^{nc}$, where nc = number of object categories = 43) generated by the semantic backbone into the SAA and CCA Modules. The collection of enhanced feature maps output by our two novel modules are processed by the UPerNet [41] decoder network. Specifically, with output features produced by SAA and CCA Modules, the decoder which adopts the Feature Pyramid Network structure was configured in accordance with the original paper: Internal Channel Number = $\{128, 256, 512, 1024\}$, Linear Layer Dimension = 512, Pooling Scales = $\{1, 2, 3, 6\}$.

## 4.1    Backbone Networks

The backbone networks complement each other in that we hope to explicitly leverage spatial and semantic features. The spatial-wise attention in the SegFormer backbone offers insight into each object's geographic information along with corresponding proximity. This considers that objects can arise in different regions in the picture under various types of circumstances. For example, glass windows of commercial buildings that situate in different corners of the scene. Despite the spatial distance, they have hidden dependencies and should belong to the same category. Meanwhile, glass surfaces can appear in the form of a 'glass door' in a corner and a 'glass table' in another. Given the different forms of existence on top of varying physical shapes, we need to devise the conceptual meanings such that the model can better differentiate the semantic categories while correlating the implicit relationships among objects. The SegFormer [15] backbone comes into play with its capacity to capture long-range dependencies and correlate spatial features with attention. With its lightweight capacity, the ResNet [25] semantic backbone serves as an auxiliary semantic context aggregator. Concretely, the integration of both paths will enable the differentiation of object representations and correlation on object dependency. According to empirical results (Section 5.3), we segregate the backbone feature layers in terms of low-level $f_l$ for $l = \{1, 2, 3\}$ and high-level $f_h$ for $h = 4$ features respectively for SSA and CCA Modules. The low-level features are used for context and object differentiation as they retain high-resolution spatial context and thus are effective for fine-grained details. The high-level features embedded with richer and abstract context information are used for correlations on category-specific knowledge.

## 4.2    Scene Aware Activation (SAA) Module

Inspired by [42], the information contained in feature maps from each layer can be further reinforced through selection and activation operations. Compared to [42], which only considers generic convo-

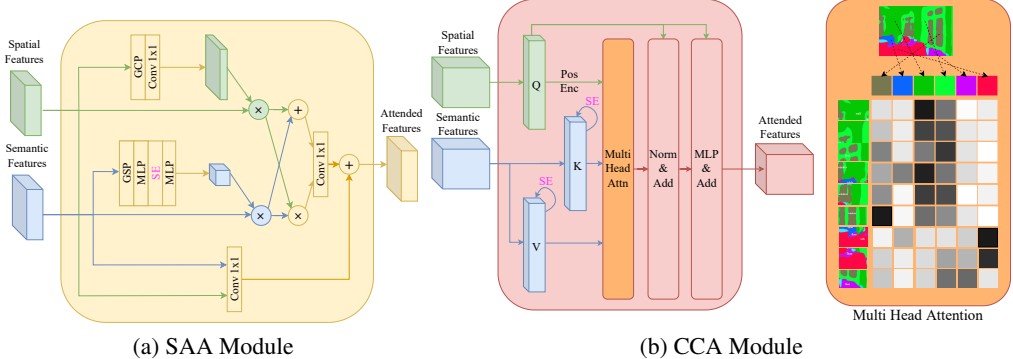

(a) SAA Module          (b) CCA Module

Figure 5: The SAA Module takes low-level backbone features to activate and discriminate object meanings. Higher-level features are used for contextual information correlation.

lutional layers, we decouple the enhancement process into respective spatial and semantic paths to suit our contextual learning settings.[3]

$$f_{sp} \in \mathbb{R}^{H \times W \times C} \rightarrow f'_{sp} \in \mathbb{R}^{H \times W \times 1}$$

$$f_{se} \in \mathbb{R}^{H \times W \times C} \rightarrow f'_{se} \in \mathbb{R}^{nc \times C}$$

$$Activation(f_{sp}, f_{se}; f_{sem\_encod}) = (f_{sp} \times f'_{sp}) \circledast (f_{se} \times (f'_{se} + f_{sem\_encod})')$$

Given the spatial backbone features $f_{sp}$, it would be first compressed into $f'_{sp}$ using Global Channel Pooling ('GCP') operations with respect to channel dimension before getting projected back onto $f_{sp}$. For $f_{se}$, it would be compressed spatially using Global Spatial Pooling ('GSP') before being flattened (i.e. $f_{se} \in \mathbb{R}^{HW \times C}$). It then will be transformed into $f'_{se}$ using a 'Multi-Layer Perception' ('MLP') operation, with the number of channels changing from originally HW to $nc = 43$ (number of classes). After that, these transformed features will be integrated with the semantic encodings from the semantic backbone. This insertion of contextual knowledge, indicated by the 'Semantic Encoding' ('SE') stage, colored in pink in Figure 5, fosters the subsequent refinement modules to be aware of the object correlations for better target localization. Note that the resultant $(f'_{se} + f_{sem\_encod})' \in \mathbb{R}^{nc \times C}$ will be re-converted back into $\mathbb{R}^{HW \times C}$ through MLP and reshaped into the same shape as $f_{se} \in \mathbb{R}^{H \times W \times C}$ before the final projection. Lastly, $\circledast$ indicates the arithmetic fusion operations to integrate the processed spatial and semantic features, as detailed in Figure 5(a).

### 4.3 Context Correlation Attention (CCA) Module

Witnessing the success of ViT [32], the attention mechanism significantly promotes the modeling efficiency on long-range dependency, enabling objects in any region to be thoroughly analyzed. Existing methods generate the (query, key, value) triplets from single input to achieve self-attention. We propose to bifurcate the attention procedure with respect to spatial and semantic features to model the correlation between objects of different categories and their corresponding locations.

$$f_{sp} \in \mathbb{R}^{H \times W \times C} \rightarrow Q \in \mathbb{R}^{HW \times C}$$

$$f_{se} \in \mathbb{R}^{H \times W \times C} \rightarrow [K \in \mathbb{R}^{nc \times C}; V \in \mathbb{R}^{nc \times C}]$$

$$Attention(Q, K, V; f_{sem\_encod}) = \text{softmax}(\frac{Q(K + f_{sem\_encod})^{\top}}{\sqrt{d_k}})(V + f_{sem\_encod})$$

Similar to the preprocessing performed for SAA Module, spatial and semantic features $f_{sp}$ and $f_{se}$ are first converted to {Query, Key, Value} triplet through flattening and channel compression. Specifically, the 'Key' and 'Value' features will go through transformation[17] (indicated by the self-loop in blue). Semantic encodings are then reinforced into these transformed bottleneck layer features before conducting feature attention modeling.

---

[3]Formulaic details have been omitted for brevity.

Table 2: Evaluation results on GDD and GSD.

| Dataset | | GDD | | | | GSD | | | |
|---|---|---|---|---|---|---|---|---|---|
| Methods | Venue | IOU↑ | $F_\beta$ ↑ | MAE↓ | BER↓ | IOU↑ | $F_\beta$ ↑ | MAE↓ | BER↓ |
| PSPNet [48] | CVPR 2017 | 0.792 | 0.875 | 0.132 | 11.51 | 0.703 | 0.834 | 0.110 | 10.66 |
| BDRAR [49] | ECCV 2018 | 0.800 | 0.908 | 0.098 | 9.87 | 0.759 | 0.860 | 0.081 | 8.61 |
| BASNet [50] | ICCV 2020 | 0.808 | 0.891 | 0.106 | 9.37 | 0.698 | 0.808 | 0.106 | 13.54 |
| MINet [51] | CVPR 2020 | 0.844 | 0.919 | 0.077 | 7.40 | 0.773 | 0.879 | 0.077 | 9.54 |
| GateNet [52] | ECCV 2020 | 0.817 | 0.931 | 0.073 | 8.84 | 0.689 | 0.898 | 0.073 | 10.12 |
| MirrorNet [27] | ICCV 2019 | 0.851 | 0.903 | 0.083 | 7.67 | 0.742 | 0.828 | 0.090 | 10.76 |
| PMD [53] | CVPR 2020 | 0.870 | 0.930 | 0.067 | 6.17 | 0.817 | 0.890 | 0.061 | 6.74 |
| GDNet [3] | CVPR 2020 | 0.876 | 0.937 | 0.063 | 5.62 | 0.790 | 0.869 | 0.069 | 7.72 |
| GlassNet [4] | CVPR 2021 | 0.881 | 0.932 | 0.059 | 5.71 | 0.836 | 0.901 | 0.055 | 6.12 |
| Ours | | **0.908** | **0.950** | **0.045** | **4.34** | **0.856** | **0.920** | **0.044** | **5.60** |

# 5 Experiments

## 5.1 Implementations

Specifically, the SegFormer backbone adopted variation B5 of Mix Transformer encoders (MiT-B5) The model is coupled with pre-trained weights for the purpose of transfer learning. The ResNet backbone is based on PyTorch's DeeplabV3-ResNet50 model [25] pre-trained on COCO train2017 [43] with only 21 categories from Pascal VOC [44]. We further trained the model using our GSD-S dataset to introduce a more diverse set of object categories for better semantic extraction capacity. Note that the semantic backbone after pre-training is fixed and isolated from subsequent training processes for better glass surface detection, lest additional information would distort the learned semantic representations. Kaiming uniform initialization [45] is used before the model was trained on an NVidia RTX 2080Ti GPU. The input data is first uniformly resized to the size of $384 \times 384$ before applying normalization. A joint loss, which is a combination of binary cross entropy and Lovász-Softmax loss [46], was used to supervise the intermediate feature maps (i.e. layers 2 and 4) and final output. The prediction evaluation is accompanied by Fully Connected Conditional Random Fields (CRF) [47] technique for binarization refinement. The evaluation metrics include intersection over union (IoU), Mean Absolute Error (MAE), maximum F-measure ($F_\beta$), and balance error rate (BER).

## 5.2 Comparisons

Table 3: Evaluation results on GSD-S.

| Methods | Venue | IOU↑ | $F_\beta$ ↑ | MAE↓ | BER↓ |
|---|---|---|---|---|---|
| PSPNet [48] | CVPR 2017 | 0.560 | 0.679 | 0.093 | 13.40 |
| DeepLabV3+ [54] | CVPR 2018 | 0.557 | 0.671 | 0.100 | 13.11 |
| PSANet [55] | ECCV 2018 | 0.550 | 0.656 | 0.104 | 12.61 |
| DANet [56] | CVPR 2019 | 0.543 | 0.673 | 0.098 | 14.78 |
| SCA-SOD [57] | ICCV 2021 | 0.558 | 0.689 | 0.087 | 15.03 |
| SETR [58] | CVPR 2021 | 0.567 | 0.679 | 0.086 | 13.25 |
| Segmenter [59] | ICCV 2021 | 0.536 | 0.645 | 0.101 | 14.02 |
| Swin [60] | ICCV 2021 | 0.596 | 0.702 | 0.082 | 11.34 |
| ViT [61] | ICLR 2021 | 0.562 | 0.693 | 0.087 | 14.72 |
| SegFormer [15] | NeurIPS 2021 | 0.547 | 0.683 | 0.094 | 15.15 |
| Twins [62] | NeurIPS 2021 | 0.590 | 0.703 | 0.084 | 12.43 |
| GDNet [3] | CVPR 2020 | 0.529 | 0.642 | 0.101 | 18.17 |
| GlassNet [4] | CVPR 2021 | 0.721 | 0.821 | 0.061 | 10.02 |
| Ours | | **0.753** | **0.860** | **0.035** | **9.26** |

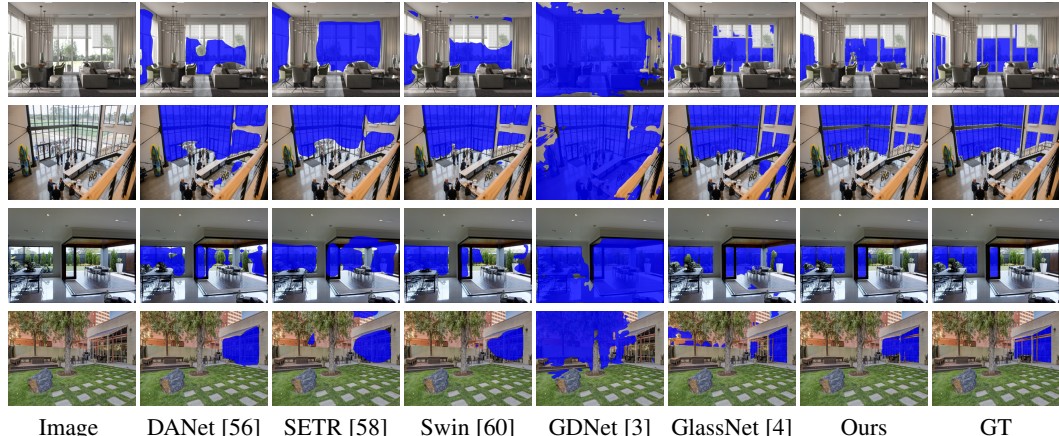

| Image | DANet [56] | SETR [58] | Swin [60] | GDNet [3] | GlassNet [4] | Ours | GT |

Figure 6: Existing methods tend to mis-detect non-glass regions due to some distractions (e.g., window blinder in the $1^{st}$ row and open area in the $3^{rd}$ and $4^{th}$ rows). Our model can accurately determine the glass regions by learning their correlation with the surrounding semantics.

We evaluated GlassSemNet against 13 other latest methods, including PSPNet [48], DeepLabV3+ [63], PSANet [55], DANet [56] for generic semantic segmentation, and SCA-SOD [10] for Salient Object Detection; recent avant-garde models that utilize transformer technique such as SETR [58], Segmenter [59], Swin [60], ViT [32], SegFormer [15], Twins [62]; and glass surface detection models, GDNet [3] and GlassNet [4]. These methods are re-trained on GDD, GSD, and GSD-S, following the default training settings stated in the original papers. Table 2 and Table 3 outlines the quantitative performance on the three glass detection datasets concerning the four evaluation metrics, which shows that GlassSemNet gives a major performance increase compared to most models. Compared to the second best model i.e. GlassNet [4], GlassSemNet surpasses with an improvement of 4.44%, 4.75%, 42.6% and 7.58% respectively for IOU, $F_\beta$, MAE, and BER on GSD-S. On the other hand, the significant disparity illustrates the challenging nature of our dataset since GSD-S has a relatively small area ratio for glass surfaces, thus giving more room for a diversified set of additional objects that offer richer semantic context. While previous methods, e.g. GDNet, are not completely catered for such assorted scenarios, GlassNet stands out with its Rich Context Aggregation Module. However, with the assistance of an understanding of intricate scene context, the performance of GlassSemNet is further elevated.

Figure 6 shows the qualitative comparisons of our method with the state-of-the-art. Existing methods that rely on boundary features [3] and contextual contrasts [4] often find the scene confusing with the presence of diverse object categories. In contrast, GlassSemNet can handle these complex cases and correctly segment the glass regions. We also conducted additional tests on real-world outdoor scenes for robustness in terms of generalization, which proved the effectiveness of our model.

### 5.3 Ablation Study

Ablation study was conducted to validate the contribution of each module, as detailed in Tables 4-8. Note that 'BB' refers to the backbone features $f_{sp}$ and $f_{se}$ while 'SE' refers to the 'Semantic Encoding' insertion into the backbone features as illustrated in Figure 5.

**Split Level and Attention Encoding Formulation.** We started by tuning the threshold of low- and high-level feature segregation (Versions A - C, D - Final in Table 4). For instance, we changed the module inputs.

from     $SSA(f_{low})$ where low = {1, 2} and $CCA(f_{high})$ where high = {3, 4} (Ver. B)

to     $SSA(f_{low})$ where low = {1, 2, 3} and $CCA(f_{high})$ where high = {4} (Ver. C).

A mild degradation is observed in IOU, $F_\beta$, and MAE. On the other hand, we compared the two options of embedding vectors (Q, K, V) assignment. The initial trial designates spatial features to be {Query, Value} embeddings and semantic backbone features to be "Key" embeddings (Ver. A

Table 4: Ablation study (Split and QKV).

| Version | Split | SSA | CCA (Spatial\|Semantic) | IOU↑ | $F_\beta$ ↑ | MAE↓ | BER↓ |
|---------|-------|-----|------------------------|------|------|------|------|
| A | 1\|234 | BB + SE | BB + SE (QV\|K) | 0.748 | 0.853 | 0.0360 | 9.66 |
| B | 12\|34 | BB + SE | BB + SE (QV\|K) | 0.750 | 0.850 | 0.0359 | 9.39 |
| C | 123\|4 | BB + SE | BB + SE (QV\|K) | 0.749 | 0.847 | 0.0366 | 9.27 |
| D | 1\|234 | BB + SE | BB + SE (Q\|KV) | 0.747 | 0.853 | **0.0346** | 9.62 |
| E | 12\|34 | BB + SE | BB + SE (Q\|KV) | 0.749 | 0.852 | 0.0358 | 9.41 |
| Final | 123\|4 | BB + SE | BB + SE (Q\|KV) | **0.753** | **0.860** | 0.0351 | **9.26** |

- C). The roles are then switched in the subsequent trials (Version D - Final), where performance gains are observed. We deduced that after the correlation mapping between Q and K, using semantic backbone features as V allows a more variegated query space than that by spatial features, which only has information in terms of separate patches over the scene.

**Semantic Encoding Enhancement.** Category-specific information was asserted iteratively into SSA and CCA Modules (Table 5). Compared to other partial insertions (Ver. A - C), The positive effect is backed by apparent enhancement when the encodings were applied on both modules (Ver. D).

Table 5: Ablation study (Semantic Encoding).

| Version | Split | SSA | CCA (Spatial\|Semantic) | IOU↑ | $F_\beta$ ↑ | MAE↓ | BER↓ |
|---------|-------|-----|------------------------|------|------|------|------|
| A | 123\|4 | BB | BB (Q\|KV) | 0.751 | 0.856 | **0.0350** | 9.30 |
| B | 123\|4 | BB + SE | BB (Q\|KV) | 0.746 | 0.849 | 0.0363 | 9.60 |
| C | 123\|4 | BB | BB + SE (Q\|KV) | 0.747 | 0.854 | 0.0361 | 9.84 |
| Final | 123\|4 | BB + SE | BB + SE (Q\|KV) | **0.753** | **0.860** | 0.0351 | **9.26** |

**Proposed Modules.** As manifested by the results upon module removals (Table 6), the results demonstrate that the mutual presence of both SSA and CAA Modules can offer a certain advancement on all metrics compared to the cases when either one was missing. This confirms the significance of the SSA Module in object characteristic differentiation, as well as that of the CAA Module on context correlation.

Table 6: Ablation study (SAA and CCA).

| Version | Split | SSA | CCA (Spatial\|Semantic) | IOU↑ | $F_\beta$ ↑ | MAE↓ | BER↓ |
|---------|-------|-----|------------------------|------|------|------|------|
| A | 123\|4 | ✗ | BB + SE (Q\|KV) | 0.747 | 0.854 | 0.0364 | 9.40 |
| B | 123\|4 | BB + SE | ✗ | 0.747 | 0.855 | 0.0361 | 9.88 |
| Final | 123\|4 | BB + SE | BB + SE (Q\|KV) | **0.753** | **0.860** | **0.0351** | **9.26** |

**Semantic Backbone.** To verify the effectiveness of our GSD-S Dataset, we vary the configurations on GSD-S fine-tuning ("train") and parameter update ("fix") of the semantic backbone network. In general, as can be seen from Table 7, fine-tuning on GSD-S (Ver. B and C) does indeed offer an incremental performance upgrade, regardless of the parameter update fixing. However, fixing the gradient resulted in a comprehensive boost across all the evaluation metrics. This is in accordance with our hypothesis that semantic priors from pre-training can be better preserved from information distortion by downstream tuning and thus provides more accurate and insightful contextual meanings.

**Cross-Dataset Analysis.** To illustrate the effectiveness of our proposed dataset GSD-S for model training, we conducted cross-dataset analysis[64] on GSD [4] (Table 8) by training on out-of-distribution data, i.e. GDD [3] and GSD-S. The result showed that the model trained on GSD-S generalizes better than that on GDD.

## 5.4 Limitation

Our model would have constrained performance under some challenging circumstances. For instance, in the presence of a "mirror" where there exist high-resolution reflections, scenery along with clear

Table 7: Ablation study (Semantic Backbone).

| Version | Split | SSA | CCA (Spatial\|Semantic) | IOU↑ | $F_\beta$ ↑ | MAE↓ | BER↓ |
|---------|-------|-----|-------------------------|------|-------------|------|------|
| A | 123\|4 | BB + SE | BB + SE (Q\|KV ✗ train ✗ fix) | 0.739 | 0.847 | 0.0377 | 10.06 |
| B | 123\|4 | BB + SE | BB + SE (Q\|KV ✗ train ✓ fix) | 0.744 | 0.851 | 0.0361 | 9.70 |
| C | 123\|4 | BB + SE | BB + SE (Q\|KV ✓ train ✗ fix) | 0.747 | 0.855 | 0.0356 | 9.52 |
| Final | 123\|4 | BB + SE | BB + SE (Q\|KV ✓ train ✓ fix) | **0.753** | **0.860** | **0.0351** | **9.26** |

Table 8: Ablation study (Cross Dataset).

| Train | Test | IOU↑ | $F_\beta$ ↑ | MAE↓ | BER↓ |
|-------|------|------|-------------|------|------|
| GDD | GSD | 0.701 | 0.782 | 0.129 | 11.20 |
| Glass-Seg | GSD | **0.774** | **0.882** | **0.0857** | **9.44** |

semantic context is reflected. This leads to a wrong prediction of false-positive glass surface presence inside the mirror region (shown in the left group of Figure 7), which is admittedly unsatisfactory. Moreover, small surfaces are often difficult to identify, given the limited space of image content occupied. Thus, the diminished attention makes it go unnoticed (illustrated by the glass table in the right group). This challenge applies to most models, including the semantic backbone network and our main model. In the future, it is hoped that we can integrate different detection strategies to alleviate this bottleneck.

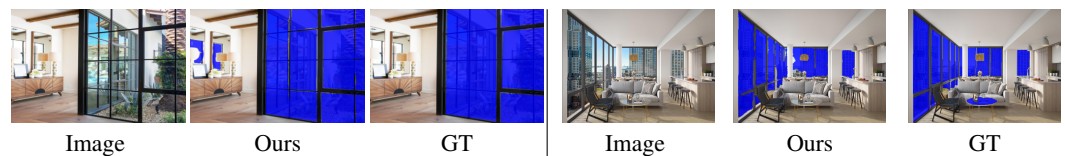

| Image | Ours | GT | Image | Ours | GT |

Figure 7: Limitations. Our method may fail to detect glass surfaces in some very challenging scenes with ambiguous visual semantics caused by mirrors or objects of small sizes.

## 6   Conclusion

In this paper, we have proposed to consider semantic knowledge in combination with spatial information to better capture the scene context as a strategic enhancement for tackling the glass surface detection problem. This comes with a meticulously constructed large-scale dataset with refined ground truth masks for both glass surface detection and semantic segmentation. Thorough experimentation demonstrates the capability of the SAA Module on object characteristic differentiation and the effectiveness of the CCA Module on context correlation. Our experiments show that the proposed model sets new benchmark records on all existing glass detection datasets, including GDD [3], GSD [4], and our GSD-S.

**Acknowledgements** This work is partially supported by two SRG grants from City University of Hong Kong (Ref: 7005674 and 7005843).

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
