# Supplementary Material: Exploiting Semantic Relations for Glass Surface Detection

**Jiaying Lin**[*]    **Yuen-Hei Yeung**[*]    **Rynson W. H. Lau**
Department of Computer Science
City University of Hong Kong
{jiayinlin5-c, yh.y}@my.cityu.edu.hk, rynson.lau@cityu.edu.hk

In this supplement, we first provide more analysis of our proposed dataset. We also provide more qualitative visual comparisons between existing state-of-the-art methods from relevant fields and our model.

## 1   Dataset Analysis

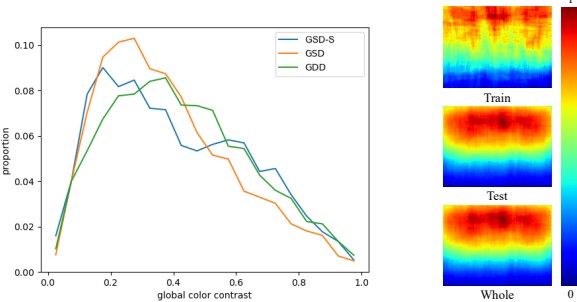

Figure 1: Statistics: (a) Color Contrast, and (b) Mask Location Distribution.

Considering the transparency nature, content inside and outside of the glass should share a high degree of semantic similarity. A lower contrast between regions decreases the saliency of particular areas and avoids causing bias to misguide model training. We measure the color contrast by computing the $\chi^2$ distance of glass and non-glass regions between RGB histograms. Figure 1a shows results compared among GDD [1] and GSD [2]. In general, the contrast values of GSD-S images concentrate in the lower quartile ($0 < \text{contrast} < 0.4$), which is comparable to GDD and GSD.

The glass location distribution is the average of all glass surface regions in the dataset. The maps in Figure 1b show that glass surfaces mainly concentrate on the top area and are consistent throughout the training and testing split. This also avoids the 'center bias' problem due to natural observation tendency.

## 2   More Experimental Results

Figures 2 and 3 show more experimental results of our method, wherein it still outperforms other state-of-the-art methods under challenging scenarios from out-of-distribution outdoor environments. Extensive testings were performed on 13 models altogether, including DeepLabV3+ [3], DPT [4], PSANet [5], PSPNet [6], ViT [7], Twins [8], SegFormer [9], Segmenter [10], DANet [11], SETR [12], Swin [13], GDNet [1] and GlassNet [14].

---

[*]Joint first authors.

36th Conference on Neural Information Processing Systems (NeurIPS 2022).

In Figure 2, rows 1 to 4 show that our method can accurately identify the glass-related objects and consequently localize the glass surfaces. Meanwhile, other models had difficulty recognizing the glass objects in the first place. Rows 5 and 6 show the robustness of GlassSemNet, that even though the glass stairs are saliently apparent, the transparent nature caused itself to blend into the surrounding scene and made it complicated for the other methods to detect. Specifically, the glass railing in row 2 (on the far right) is boundless, which went unnoticed by most models while GlassSemNet was aware of its presence. Rows 4 and 5 are the strong indicators that our model is able to infer the contextual relationships by showing that the humans in row 5 belong to the indoor area inside the building (thereby including them in the glass surface prediction mask). The street light in row 6 belongs to the outdoor area outside of the building (thereby excluding it from the prediction mask).

We can see that GlassSemNet can utilize the semantic context to infer the object relationship, which fosters more cognitive reasoning with contextual inference. Figure 3 continues to demonstrate the effectiveness of GlassSemNet under demanding scenarios.

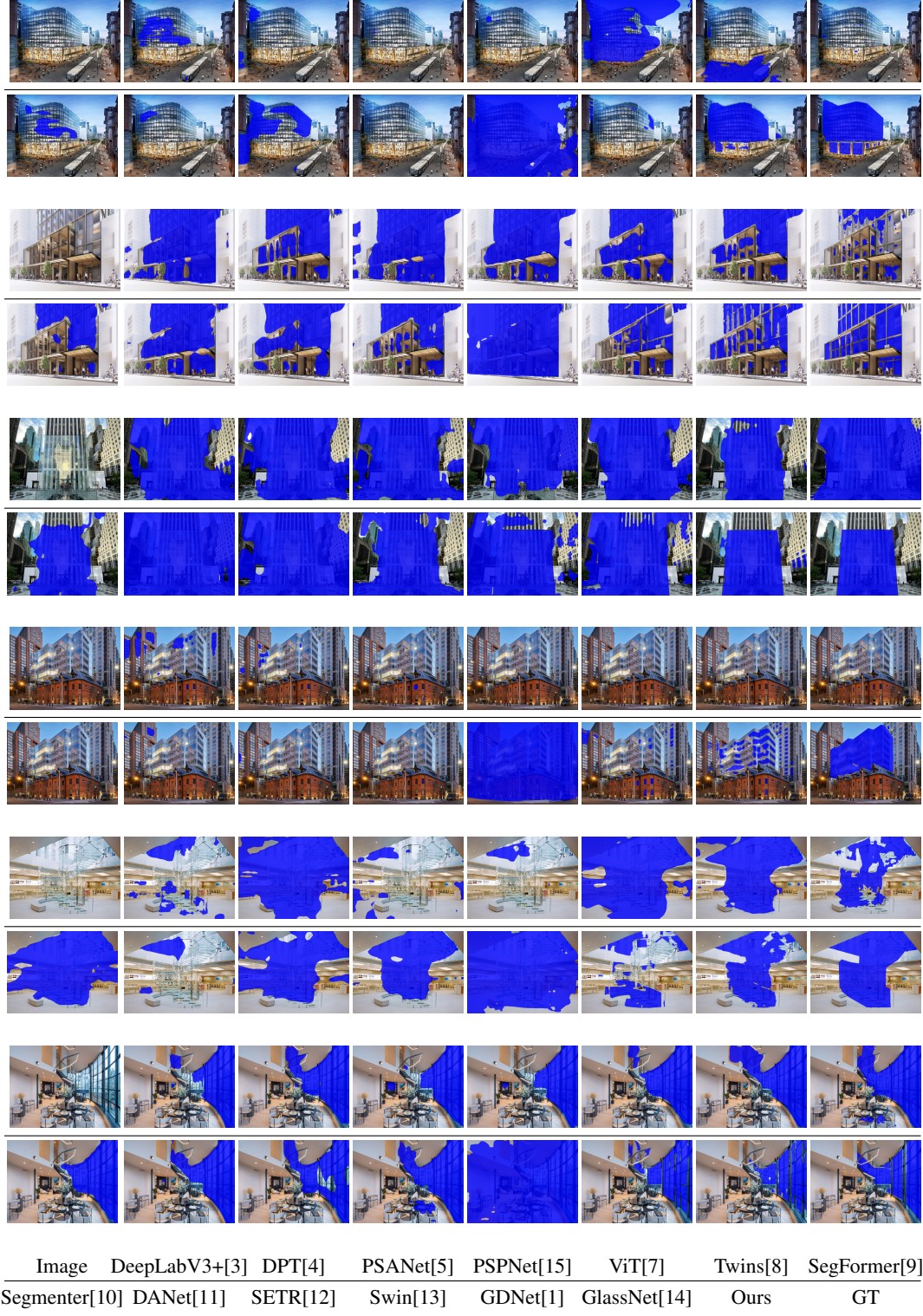

Figure 2: Visual comparisons of our method to the state-of-the-art methods.

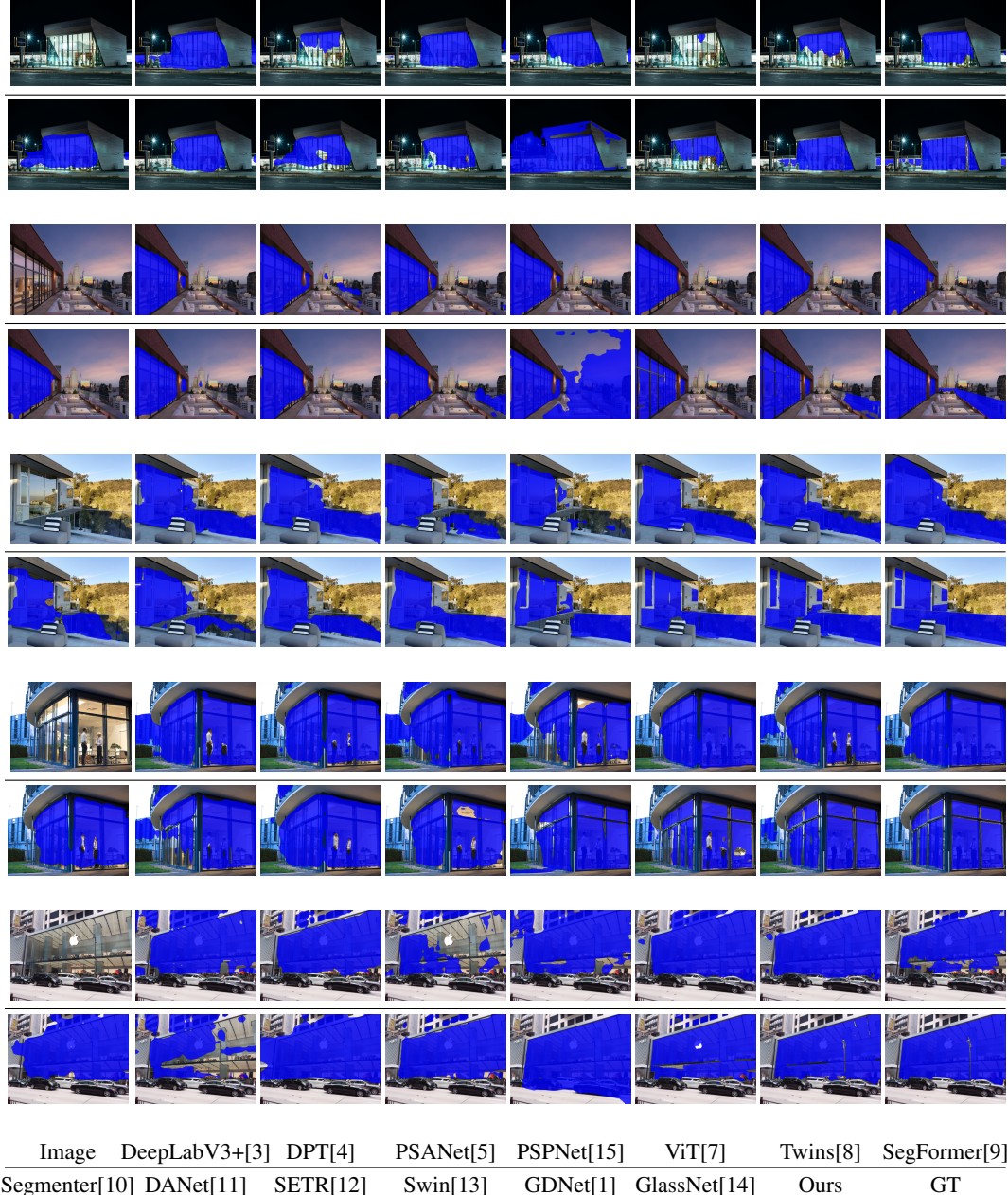

Image    DeepLabV3+[3]    DPT[4]    PSANet[5]    PSPNet[15]    ViT[7]    Twins[8]    SegFormer[9]

Segmenter[10]    DANet[11]    SETR[12]    Swin[13]    GDNet[1]    GlassNet[14]    Ours    GT

Figure 3: (Cont.) Visual comparisons of our method to the state-of-the-art methods.