# OpenReview forum: "Exploiting Semantic Relations for Glass Surface Detection"
_NeurIPS.cc/2022/Conference — NeurIPS 2022 Accept_

### Official Review · Reviewer_pjyT · 2022-07-09

**Rating:** 4
**Confidence:** 5
**Soundness:** 2 fair
**Presentation:** 2 fair
**Contribution:** 2 fair

**Summary:**

The authors observe that the glass object usually cooccurs with some specific objects. Based on the observation, the work proposes a method consisting of two main components CCA and SAA to feature selection in spatial and semantic dimension and efficiently model long range dependency.  Extensive experiments are conducted on their newly collected data to demonstrate the effectiveness of the proposed method and its superiority to the state-of-the-art methods.

**Questions:**

Please address the weaknesses mentioned above. My major concern is that the observation is specifically with glass object, thus the proposed method does not deal with challenges caused by the characteristics of glass object.

**Limitations:**

Yes

**Strengths And Weaknesses:**

Strengths
1. The performance of the proposed method is promising.
2. The collected data is useful to facilitate the research in this direction.

Weaknesses
1. The observation is common and lacks insight, thus making the work less inspiring. In addition, the observation of some object is closely related with some other objects is not only existed in glass object. For example, sea is usually closed with sand.
2. The proposed CCA and SAA modules are not specifically designed for glass object, it is more like a general way to model the correlation among objects. In this case, the methods should be evaluated in more general segmentation task, where the observed relationship is present as well.
3. Line 215, the authors mention that "We then further fine-tuned the model using our GSD-S dataset". Did all compared models do this fine-tuning? How dose it affect the model performance?

---

> ### Author Response · Authors · 2022-08-02
> **Response to Reviewer pjyT**
>
> We thank the reviewer for the valuable comments. We will address your concerns below.
>
> **Q1: Lack of insight.**
>
> A1: Our main objective is to tackle the challenge of detection of glass surfaces which typically lack outstanding visual properties such as clear object outline, color scheme, not even to mention object texture. Considering that contemporary works in such sub-domain mostly focus on low-level cues (e.g., glass object boundary and reflections off from the transparent object surfaces), we are trying to elevate the study on glass surface detection to a higher-level that takes into account abstract semantic contexts. As backed by Kaiser et al. (Object grouping based on real-world regularities facilitates perception by reducing competitive interactions in visual cortex), who discovered the ability of the visual cortex system to exploit real-world regularities to group co-occurring objects. Empirically, we can also conclude from our extensive experiments that our newly proposed method outperformed the existing STOA works who only leveraged low-level information, as these features are likely to be have been intrinsically captured by the model’s convolution operations.
>
> **Q2: The method should be evaluated in more general segmentation tasks.**
>
> A2: It is indeed an inspiring suggestion from the reviewer that this hypothesis can be further extended to the general semantic segmentation tasks, offering a much larger extent of contribution than merely on the sub-domain in glass surface detection. We are still training our method on some segmentation task but the results would not be available due to the short rebuttal period. This will be a future direction of our project exploration and we may discuss it in our final version.
>
> **Q3: Line 215, the authors mention that "We then further fine-tuned the model using our GSD-S dataset".  Did all compared models do this fine-tuning? How does it affect the model performance?**
>
> A3:
> No. As the all compared models do not explicitly exploit the semantic information in their network structure, fine-tuning them in the pre-training stage with our GSD-S dataset will not have performance gain and will even produce wrose results. For example, the best baseline method GlassNet will have a performance drop (IoU from 0.881 to 0.869, F_beta from 0.932 to 0.901, MAE from 0.059 to 0.070, BER from 5.71 to 6.02 on GDD) after fine-tuning on GSD-S dataset in the pre-training stage. It also reflects the necessity of proposing a new method with the proper usage of semantic information.
>
> As our model can effectively exploit semantic relations, using GSD-S dataset for fine-tuning in the pre-training stage can improve our model, especially in BER (from 11.03 to 10.56). For more details, please refer to the ablation study (Version 1 and 3 in Table 4).

---

> > ### Comment · Reviewer_pjyT · 2022-08-08
> > **discussion on response**
> >
> > Thanks so much for your response.
> > After reading the response, my concerns are not solved.

---

### Official Review · Reviewer_XdVk · 2022-07-11

**Rating:** 5
**Confidence:** 3
**Soundness:** 2 fair
**Presentation:** 2 fair
**Contribution:** 2 fair

**Summary:**

This paper proposed a new method for glass surface detection whose key insight is to incorporate semantic context information of the environment. Two new modules are proposed: (1) a Scene Aware Activation (SAA) Module that adaptively filters critical channels for spatial and semantic features and (2) a Context Correlation Attention (CCA) Module that learns the contextual correlations among objects both spatially and semantically. The paper also proposed a large-scale glass surface detection dataset named GSD-S containing detailed semantic annotations. Experiments show that the proposed method achieves state-of-the-art performance on multiple datasets including the proposed GSD-S dataset.

**Questions:**

- I think more analysis to justify the effectiveness of the proposed dataset would be helpful. How can one know if the proposed method is more effective? Do methods trained on the proposed dataset perform better on in-the-wild out-of-distribution data than on existing datasets?
- Does the proposed dataset contain mostly indoor scenes? How will the method perform on images of glasses outdoor?
- Although the glass co-occurs a lot with walls, etc, the contents that are seen through the glass can be more random and diverse. How robust is the method to different contents inside the glass area?
- The first sentence in the caption of Table 1 says "GSD dataset". Should it instead be "GSD-S dataset"?
- It is unclear which curve belongs to the proposed GSD-S dataset in Figure 2 (b) as "GSD-S" does not exist in the legend (I guess it is "Glass-Seseg"?). It is also not clear what GDD and GSD stand for without citations and captions. Suggest making Figure 2 (b) clearer.

Minor:
- Last sentence in Fig.3 caption "multiple stage" -> "multiple stages'.
- Seems that Fig 4. (a) "Attended Features" should instead be "Activated Features".





**Ethics Review Area:**

["I don’t know"]

**Limitations:**

Yes.

**Strengths And Weaknesses:**

Strengths:
+ The idea of exploring the semantic relationship between glass and surrounding objects to facilitate glass detection is interesting and insightful.
+ The paper created a larger dataset for glass detection with refined glass masks and semantic annotations, which will be useful for future research on this topic.
+ The paper conducted extensive experiments which are convincing. The method is compared to various kinds of baselines on multiple datasets, and the proposed method achieves very good results consistently. Very detailed ablation studies also justify the design choices well.

Weaknesses:
- I'm not an expert in this area, but I think the technical novelty of the proposed method is a little limited. The method builds upon many existing frameworks, and combines and adapts them to the glass detection problem, which makes the technical novelty incremental, although I think the insights behind the design are good.
- The method section contains too many details which are overwhelming to read. For example, in Sec 4.2, Eq (1) and Fig 4 (a) give a very detailed description of each individual operation, which kind of goes into the weeds. I think it would be better to cut some unimportant details or move them to the section of implementation details.

Given the interesting idea, the good performance, and the extensive evaluation, I currently vote for borderline acceptance despite that the technical novelty is a bit limited. I would like to see reviews from other reviewers and rebuttals from the authors to make further decisions.

---

> ### Author Response · Authors · 2022-08-02
> **Response to Reviewer XdVk**
>
> Thank you for recognizing the insight, the good performance, and the extensive evaluation of our work. We will address your concerns below.
>
> **Q1: Limited novelty although good insights behind the design.**
>
> A1: Thank you for the positive comments on the insights behind our designed method. We argue that the key novelty of our paper is not using existing frameworks (e.g., backbone network), which is a common way for transfer learning in downstream tasks. Instead, it comes from connecting these existing frameworks by the under-explored strategy that models semantic relationships between the glass objects and everyday objects, and two novel modules (e.g., the SAA module and CCA module) to capture long-range spatial and implicit semantic dependencies for glass surface detection.
>
> **Q2: Remove unimportant details.**
>
> A2: Thank you for your constructive suggestion. We will rewrite the description of the mentioned parts.
>
> **Q3: I think more analysis to justify the effectiveness of the proposed dataset would be helpful. How can one know if the proposed method is more effective? Do methods trained on the proposed dataset perform better on in-the-wild out-of-distribution data than on existing datasets?**
>
> A3: To verify the effectiveness of our dataset, we conduct cross-dataset analysis [Ref1] on existing glass surface detection datasets. (e.g., GDD, GSD and our GSD-S).
>
> Table 1 show the results of our cross-dataset analysis on the test set of GSD, i.e., training a model on one other dataset (e.g., GSD-S or GDD) and testing it on the test set of GSD. We can see that the model trained on our GSD-S will have superior performance than the one trained on GDD on the GSD test set. This indiciates our dataset is more effective.
>
>
> |                    | IoU      | $F_\beta$|   MAE  | BER     |
> | :------:           | :------: | :------: |:------:|:------: |
> | Train on GSD-S     | **0.739**     | **0.846**      |   **0.107**  |   **11.16**   |
> | Train on GDD       | 0.628      | 0.708      |   0.222  |   16.10   |
>
> Table 1. We train our method on different datasets and test it on the test set of GSD dataset to verify the effectiveness of our proposed GSD-S dataset. We can see that the model trained on our GSD-S will have superior performance than the one trained on GDD. Best results are in bold.
>
> **Q4: Does the proposed dataset contain mostly indoor scenes? How will the method perform on images of glasses outdoor?**
>
> We analyze the proposed dataset and find 4013 indoor images (88.8%) and 506 outdoor images (11.2%)  from our GSD-S. We indiviually evaluate images outdoor in the test set and find our method still performs well in outdoor scenes. (IoU: 0.744, $F_\beta$: 0.836, MAE: 0.039 BER:8.88)
>
> **Q5: Although the glass co-occurs a lot with walls, etc, the contents that are seen through the glass can be more random and diverse. How robust is the method to different contents inside the glass area?**
>
> A5: That is an interesting problem. However, to the best of our knowledge, existing datasets for glass surface detection do not contain such groups of images that have the same contents outside the glass region but varied contents inside the glass region. It will be challenging to evaluate the robustness of existing methods. It will be a good future work to construct such a dataset for robustness evaluation.
>
> **Q6: The first sentence in the caption of Table 1 says "GSD dataset". Should it instead be "GSD-S dataset"?**
>
> A6: We deeply apologize for the typo. It should be "GSD-S dataset".
>
> **Q7: It is unclear which curve belongs to the proposed GSD-S dataset in Figure 2 (b) as "GSD-S" does not exist in the legend (I guess it is "Glass-Seseg"?). It is also not clear what GDD and GSD stand for without citations and captions. Suggest making Figure 2 (b) clearer.**
>
> A7: Thank you so much for your detailed proofreading. The Glass-Seseg should be "GSD-S”. We will make a clearer Figure 2 (b) with detailed citations and captions in our revision.
>
> References:
>
> [Ref1] Antonio Torralba, Alexei A Efros, et al. Unbiased look at dataset bias. In IEEE CVPR, pages 1521–1528, 2011.

---

### Official Review · Reviewer_KsvV · 2022-07-11

**Rating:** 5
**Confidence:** 4
**Soundness:** 3 good
**Presentation:** 3 good
**Contribution:** 2 fair

**Summary:**


The paper tackles the issue of glass surface reflection from a single optical image. The authors observe that glasses are difficult to detect as isolated objects (they do not have specific radiometric or geometric properties). Consequently, the authors propose to account for the semantic context around objects, with the aim to ease the task. Concretly, the authors introduce a network architecture that encompass semantic information and low level features. A new dataset of glass is also created. Results are illustrated on a classic benchmark (Glass Detection Dataset, Mei and al. 2020), and the authors' dataset.


**Questions:**

1- I am little confused about the use of SegFormer and DeepLab, which are according to the authors, "aggregate spatial-wise object location features"  for the former, and give "intrinsic representations of each object category" for the latter. However, these two networks, to my knowledge, are two semantic segmentation networks. I might misunderstand the differences of usage of these two networks, it would be worth to clarify.

2- The training procedure is important indeed, and a few additional words on this matter would be interesting.  The authors write: "We then further fine-tuned the model using our GSD-S dataset to introduce a more diverse set of object categories for better semantic extraction capacity. Note that the semantic backbone after fine-tuning is fixed and isolated from subsequent training for glass surface detection". I do not understand why a fine tuning then a subsequent training is necessary? What is learnt at each stage and for what purpose?

3- Equations 2, describe in a very classical way the attention frameawork. What is the dimensionality of f_sp, f_se, Q,K,V.

**Ethics Review Area:**

["Responsible Research Practice (e.g., IRB, documentation, research ethics)"]

**Limitations:**

No concern with societal impact.

**Strengths And Weaknesses:**

The main idea of the authors, ie, accounting for the semantic context surrounding glass surface so as to ease the detection, is certainly interesting. Thought it has been proposed by others for other applications  (in particular for human detection),  it is particularly interesting in the current setting. To push further the idea, the problem could be formalized as establishing pair-wise dependencies between semantic objects.

The implementation of the idea makes sense to me, thought the technical description of the approach appears to me sometimes somehow vague. I wonder if this architecture could be re-used for other problems or applications, ie how much it is specific to this sole problem  at hand  or could it be useful beyond glass detection?

Results on the two datasets are encouraging. Especially the detection visualizations (figure 5, 6) show very consistent glass contours. Quantitative results are less assertive (but it is ok).

The weak point of the paper is probably the sometimes vague technical description of the approach and probably insufficient discussion about technical choices.  For example in eq.(1) ,  specifying the dimensional of the input and output tensors would  be useful. A short description of the UperNet network used for final segmentation would also have been useful. A clearer description of the learning (finetuning) procedure would also be welcome.

---

> ### Author Response · Authors · 2022-08-02
> **Response to Reviewer KsvV**
>
> Thanks a lot for your time and constructive feedback. Below we address all raised concerns with additional detailed descriptions that we will include in the final version of the paper.
>
>
> **Q1: For example in eq.(1) , specifying the dimensional of the input and output tensors would be useful.**
>
> A1: Given the backbone features $f_{sp} \in \mathbf{R}^{H \times W \times C}$. The spatial feature would be first compressed into $f_{sp}' \in \mathbf{R}^{H \times W \times 1}$ using Global Average Pooling and get multiplied back to $f_{sp}$; For $f_{se} \in \mathbf{R}^{H \times W \times C}$, it would first be flattened into $f_{se}' \in \mathbf{R}^{N \times C}$ before it goes through the semantic representation encoding process to be $\mathbf{R}^{nc \times C}$ where $ nc = 43$ (number of classes) and to $\mathbf{R}^{N \times C}$ before it is reshaped back to $\mathbf{R}^{H \times W \times C}$. The rest will be a combination of multiplication and summation operations to integrate the two input features.
>
>
> **Q2: short description of the UperNet network used for final segmentation**
>
> A2: UperNet has been fundamental in all classification and segmentation tasks, it has been utilized in influential contemporary works including Swin-Transformer. Given the inputs which come from SAA and CCA Modules, we have the following configurations for UPerNet: for Feature Pyramid Network, internal channel numbers = {128, 256, 512, 1024} and dimension for linear layer is 512. The pooling scales for the Pyramid Pooling Module follow the default settings {1, 2, 3, 6} from the original paper.
>
> **Q3: clearer description of the learning (finetuning) procedure**
>
> A3: Given that we adopted PyTorch’s pretrained DeepLab model as the semantic backbone for feature extraction. The purpose of fine-tuning on our GlassSeg is to introduce semantic contexts that contain glass surfaces for the modeling of world-interactions into DeepLab backbone. The supervision is conducted on semantic segmentation given the RGB image. After that, we would then start training the main model, including the backbone networks, SAA and CCA modules, and UperNet for glass surface detection.
>
> Minor Questions:
>
> **Q4: I am little confused about the use of SegFormer and DeepLab, which are according to the authors, "aggregate spatial-wise object location features" for the former, and give "intrinsic representations of each object category" for the latter. However, these two networks, to my knowledge, are two semantic segmentation networks. I might misunderstand the differences of usage of these two networks, it would be worth to clarify.**
>
> A4: The main reason to employ SegFormer for spatial dimension is due to its capability of capturing long-range dependencies. Under this set up, spatial features in every corner of the image can be attended and correlated. A further theoretical support for this point can be found in [Ref1], which shows vision transformers (e.g., Segformer) retain more spatial information than ResNet.
> On the other hand, for ResNet backbone from DeepLabV3-ResNet50 was adopted for semantic feature extraction due to the lightweight capacity, it serves as an auxiliary semantic context aggregator while introducing insignificant compute requirements.
>
>
> **Q5: The training procedure is important indeed, and a few additional words on this matter would be interesting. The authors write: "We then further fine-tuned the model using our GSD-S dataset to introduce a more diverse set of object categories for better semantic extraction capacity. Note that the semantic backbone after fine-tuning is fixed and isolated from subsequent training for glass surface detection". I do not understand why a fine tuning then a subsequent training is necessary? What is learnt at each stage and for what purpose?**
>
> A5: Please refer to A3.
>
> **Q6: Equations 2, describe in a very classical way the attention framework. What is the dimensionality of f_sp, f_se, Q,K,V.**
>
> A6: $f_{sp}$ and $f_{se}$ both were $\mathbf{R}^{H \times W \times C}$. Similarly to CCA module, attributes K and VI in particular $\mathbf{}^{H \times W \times C}$. The attended features Q, K, V respectively by SAA module would then become $ \mathbf{R}^{N \times C}$ except that semantic features associated with K and V attributes would also go through `Squeeze and Excitation block' to select the most information feature channels in semantic backbone features. Similar to that in  representation encoding in CCA module.
>
> Reference:
>
> [Ref1] Raghu, Maithra, et al. "Do vision transformers see like convolutional neural networks?." Advances in Neural Information Processing Systems 34 (2021): 12116-12128.

---

### Official Review · Reviewer_yyRs · 2022-07-12

**Rating:** 6
**Confidence:** 4
**Soundness:** 3 good
**Presentation:** 2 fair
**Contribution:** 3 good

**Summary:**

The paper addresses the issue of segmenting glass surfaces in an image. The novel contribution is to use semantic context more explicitly in the overall network system design, as often there are no visual features with which to identify glass. The authors also contribute an extensive new data set with carefully labeled glass areas that includes many images from standard data segmentation data sets. The new network design outperforms the state of the art methods.

**Questions:**

Did the authors find any situations where the context fooled the network into thinking glass existed when it did not?

**Limitations:**

The authors note the limitation created by mirrors or mirror-like surfaces. There are no examples shown where context fools the system into thinking glass is present when it is not (e.g. an open window).

**Strengths And Weaknesses:**

The network incorporates both a spatial and a semantic (area+category) based backbone, followed by two modules that combine inputs from both backbones using different architectures. The ablation study shows that both modules are important for performance.

The ablation study seems comprehensive, and it includes how best to assign the query/value/key concepts in the context module.

The network design outperforms existing state-of-the-art networks on both prior data sets and the new contributed data set.

The authors are providing a new and useful data set for glass detection.

Figure 3 has space for more annotation of the components. All component blocks should have a label, and the reader should not have to go back to the text to figure out what the components are (for example, if Upernet is used for the decoder, that should be clearly labeled in figure 3.

The paper needs editing for minor details.  For example, in the caption for figure 3

"The CCA Module is positioned at higher level in the encoder component to inference the relationships between contextual meanings and locations of objects. Features from multiple stage are aggregated by decoder to produce output map." -> "The CCA Module is positioned at a higher level in the encoder component to infer the relationships between contextual meanings and locations of objects. Features from multiple stages are aggregated by the decoder to produce the output map."

---

> ### Author Response · Authors · 2022-08-02
> **Response to Reviewer yyRs**
>
> We thank the reviewer for the positive comments and valuable feedback.
>
> **Q1: The illustration and caption of Figure 3 should be improved.**
>
> A1: Thank you so much for your suggestions. We will revise it in our revision.
>
> **Q2: Did the authors find any situations where the context fooled the network into thinking glass existed when it did not?**
>
> A2: Yes. Due to the limited space, we do not include the mentioned examples in our initial submission. We will add them to our final version.

---

### Author Response · Authors · 2022-08-02
**Response to All Reviewers**

We would like to thank the reviewers for the evaluation of our manuscript for the positive feedback: interesting idea (Reviewer KsvV, Reviewer XdVk), proposed useful dataset (Reviewer yyRs, Reviewer XdVk, Reviewer pjyT), rational design (Reviewer XdVk), thorough and convincing experiments (Reviewer yyRs, Reviewer XdVk), impressive performance (**ALL** Reviewers including Reviewer yyRs, Reviewer KsvV, Reviewer XdVk, Reviewer pjyT).

We will update the draft according to the suggestions on the writing and figures. For specific questions of each reviewer, we answer them in the reply to each reviewer.

---

### Meta-Review · Area_Chair_p6gH · 2022-08-22

**Recommendation:** Accept
**Confidence:** Certain

**Metareview:**

Post rebuttal, three out of four reviewers were in favor of acceptance. The hold-out reviewer pjyT was primarily concerned with the fact that the method isn't specifically tailored to the task (in the sense that other domains have similar features) and therefore the method ought to be tested in more settings. The AC understands pjyT's perspective in the sense that deep problem-specific understanding of a domain often drives methodological contributions. However, the AC does not find this to be a convincing argument for rejecting the paper and thinks the work provides enough of a strong contribution. Accordingly, the AC recommends accepting the paper.

**Award:**

No

---

### Decision · Program_Chairs · 2022-09-14

Accept